# The Negative Perceptions of Apartment Culture as Represented in Korean Films during the 1970s–1990s

**Guen-Jong Moon**

Department of Architecture, Keimyung University, Daegu 42601, Korea; gjmoon@kmu.ac.kr

**Abstract:** Popular films, which are cultural products, inevitably reflect the social and architectural culture of the time and the thoughts and interests of the public. This study analyzes the negative perceptions of apartment culture to verify how the negative characteristics of apartment housing were recognized by the general Korean public in a socio-cultural manner. For the analysis, a pool of artistically and publicly renowned Korean films between 1970 and 1999 was constructed. Through the scenes and their respective scripts, the characters, stories, cinematic messages, and architectural spaces were analyzed. The 1970s and 1980s films shed light on the large-scale, uniformly developed apartment complexes to reveal apartments as lonely, anonymous, closed spaces of the urban middle class. During the 1980s–1990s, the negative aspects of apartment developments were highlighted. These include a loss of place and memory, the disintegration of family, the deepening of relative poverty, and standardized desolated scenery. Negative perceptions toward apartments intensified in the 1990s to reveal a lack of communication between neighbors, externality, misunderstanding, and distrust. By diagnosing the Korean public's negative view of apartments, this study will help find a better housing culture and the positive sustainability of apartments.

**Keywords:** apartment; culture; negative perceptions; Korean film; 1970s–1990s

## 1. Introduction

In 2018, there were 10,826,044 apartments out of 17,633,327 houses (61.40%), while there were 3,948,984 detached houses (22.40%) in the same year [1]. As such, apartments in Korea, which are representative of residential styles, have so far formed many positive perceptions, such as being convenient and safe to live in. There are also many negative perceptions, such as them being enclosed spaces, symbols of anonymity, objects of speculation, inductions of hierarchical differentiation, and formations of desolate city landscapes. In short, apartments have grown with ambivalent and complex characteristics as both the objects of envy and objects of resentment simultaneously. Despite the various negative perceptions, there seem to be no new types of housing alternatives available except for apartments. The mass construction of apartments is currently ongoing. This paper aims to examine the negative aspects of apartments from the public's perspective with the sustainability of the housing culture in mind for apartments that occupy such an important position within Korean society.

We examine the negative perceptions of apartment culture through the analysis of Korean films between 1970 and 1999. It is assumed that these films, as cultural products, reflect the social and architectural backgrounds alongside the interests of the general public. We attempt to explore the negative aspects of the apartment housing culture based on visual scenes and their meanings in popular films. Therefore, this paper combines the perspectives of humanities and architecture through the study of these films to explore the apartment culture from a new viewpoint (rather than from an architectural expert) by using the perspective of the general public, including the filmmakers and writers. This approach is considered different from the existing architectural studies from the literature

and print media. However, on the screen, exaggeration and distortion of actual building spaces or residential cultures can occur. Therefore, this study is conducted using appropriate considerations.

Examining the negative perceptions of apartment culture means looking at the social and cultural aspects of how the public perceives the negative characteristics of the apartment housing culture at that time. By diagnosing the negative characteristics revealed through the analysis, this study will help to find better apartment housing culture and positive sustainability in Korea.

## 2. Literature Review

### 2.1. Films that Reflect Architecture and the Housing Culture

Film is a matrix of contemporary image culture, and has become an independent study field that assembles social activities and languages to become an industry within itself [2]. As a subject of cultural study, film is investigated as a cultural product and social activity. Film is valuable because it shows the systems and processes of human culture. After all, understanding a film is not necessarily an aesthetic act. It is a social act that mobilizes the whole system of meaning in culture.

Moreover, film has much in common with architecture. A director expresses urban landscapes, buildings, and interior spaces through their lens, and the public recognizes and enjoys them. We can refer to this type of process as an architectural activity alike general architectural design processes [3]. A film director or writer tends to reflect the public's general consciousness and emotion about architecture at a given time. This is due to the fact that given the nature of mass media, which must target a large number of unspecified and widespread masses, film and television must take into account their diverse tastes and needs [4]. In a similar context, Steven Jacobs said, "All kinds of vernacular architecture represented on the screen expresses opinions and feelings that people nourished about small, big, beautiful, and ugly buildings. Screen architecture demonstrates the ways that people have put meanings on the notions of the home, domestic culture, public spaces, landscapes, monuments, the differences between inside and outside, and so forth" [5] (p. 10). Moreover, Beatriz Colomina suggested that architecture could be regarded as a form of mass media when she stated that "A building should be understood from different points of view such as a painting, photo, writing, films, and advertisements; not only because these are the media in which we more often encounter it, but because the building is a mechanism of representation in its own right" [6].

### 2.2. Apartments in Korean Films during the 1930s–1960s

The 1930s and 1940s saw the initial introduction of apartments, and apartments were represented in film as urban, modern, and Western-style housing. Regarding the apartments of the 1950s and 1960s, positive images such as abundance, convenience, and cleanliness were added to the previous modern and Western images. However, they also appeared as a poor residential environment for ordinary people and the poor [7]. Moreover, in the uneasy period between post-war Korea and its rapid modernization, the apartment emerged as a space for the anxious modern woman, and in some films, apartments were experimentally treated as such. As such, apartments in this period appeared to contain a variety of meanings.

One of the things left in Korean society after the U.S. military's rule and the Korean War was extreme poverty, and the apartments of the 1960s, including the Citizen's apartments, were built on the slopes of low-income housing, which was of low quality. As such, the apartment as a poor living environment among the aforementioned content can be interpreted as a natural phenomenon that reflected the social situation of the time. However, the modern and Western image of the apartment was due to the public's enormous envy and curiosity about the Western lifestyle in the dark period of Korean society. In short, positive perceptions of apartments as poor living spaces were expressed through films, but the negative perceptions of an apartment itself were not revealed until the late 1960s.

## 3. Research Scope and Methods

This study explores the instances where Korean apartments appeared in films or scenarios between 1970 and 1999. Although all scenarios or films in which apartments appear could be selected at this time (considering that there are approximately 80 films released in Korea each year), it was necessary to reasonably reduce the population of films for selection before selecting the films for analysis. Therefore, the films chosen for this study had already been selected as distinguished entrants in cultural collections: The "Korean scenario collection," a series of films selected and published by the Korean Film Council (the initial period began around 2006), the "Korean film 70 years collection 200" (the initial period began around 1989) [8], and the "Korean film collection 100", as selected by the Korean Film Archive in 2006 (the initial period began around 1996). The film selection criteria of these three populations were award-winning or nominated films at domestic and international film awards, films with more than one hundred thousand viewers, and films significant in the understanding of Korean society and culture, amongst others, and thus were deemed to be of high quality and popular, relatively reflecting the social and housing culture and public interests of the time. Among the films in the abovementioned collections, a total of 180 works from 1970–1999 were reviewed, and 62 films and scenarios in which an apartment appeared were identified. Through these 62 films, the scenes and their respective scripts were analyzed. It was expected that each scene of the films would play visually meaningful roles. An analysis of the scripts would help understand the characters, stories, descriptions of architectural spaces, and messages of the films. The perceptions of apartments were grouped into five categories. The first concerned an apartment as an object of admiration or a happy residential space intensively in the 1980s and 1990s. The second was an apartment as a space combined with femininity. This appears consistently from the Korean War until today. The third was an apartment as an everyday space for an ordinary middle-class family. This did not appear until the 1970s, but frequently appeared in the 1980s. Fourth, an apartment as a residence of a non-mainstream member of society was also revealed. Fifth, perceptions such as loss of place, lack of communication, distrust, and anxiety, in addition to loneliness, anonymity, and closure, emerged in the 1970s and has been intensifying.

The temporal scope was limited between 1970 and 1999. In Korea, the construction of apartments started with the Toyoda apartment in Seodaemun-gu, Seoul, in 1930, followed by the Haengchon apartments (3 buildings and 48 units) in Jongno-gu, Seoul, in 1956. The Seoul Mapo apartments (1962) became known as the beginning of the emergence of apartment complexes. Meanwhile, the mass production and distribution of Korean apartments began with the Citizen's apartments in 1969 (426 buildings, 16,963 units) and the Banpo Housing Complex in the early 1970s in Seoul. Further, the Banpo Housing Complex was the starting point of the full-scale realization of the middle class, along with the Han River mansions (1970) and the Yeouido Sibeom apartments (1971). Thus, the temporal range of the 1970s–1990s was the period after the full-scale mass spread of apartments and the full-scale realization of the middle class.

## 4. Results: Negative Perceptions of Apartments in Films of the 1970s–1990s

### 4.1. Lonely and Anonymous Spaces of the Urban Middle Class

The housing construction boom occurred in the 1970s due to demand, which resulted from speculations concerning a surge in housing prices. In addition, this period of rapid development of apartment complexes was a period in which the conventional practice of universal planning began. In particular, the method of close and exclusive planning for the surrounding street spaces by intensively arranging residential centers, such as shopping areas, within the housing complex and surrounding the housing complex with a fence started to become conventional practices [9]. Almost as if reflecting this phenomenon, the apartment culture in the films of the 1970s and 1980s was expressed mainly through loneliness, alienation, emptiness, closure, anonymity, and bleakness, rather than revealing the everyday life of the urban middle class. This can be seen in films from the following films of this period.

The film "Night Voyage" (Soo-yong Kim, 1973) focuses on the monotonous daily life, loneliness, and mental wandering of the main female character. The spatial background is set in the Banpo Housing Complex, which had just been completed and had allowed residents to move in (see Figure 1). The film is set against the backdrop of Korea's modernization period, which demanded citizens' uniformity in life and production. The main female character, Hyun-joo, is a spinster who works at the bank. Her life is very regular, and she returns home right after work. She secretly lives with a man who works at the same bank. Their relationship resembles that of a bored couple. The nightlife they experience together does not provide an escape or shelter from their daily lives at work, and their nights are just as customary and numb as their days. In short, her recurring routine runs monotonously at the bank in the day and the apartment at night. Cleaning, cooking, and lovemaking in the indoor space cause her to wander and deviate. Here, the space of the apartment is the freest space, yet simultaneously represents loneliness, alienation, emptiness, lack of communication, and mental wandering.

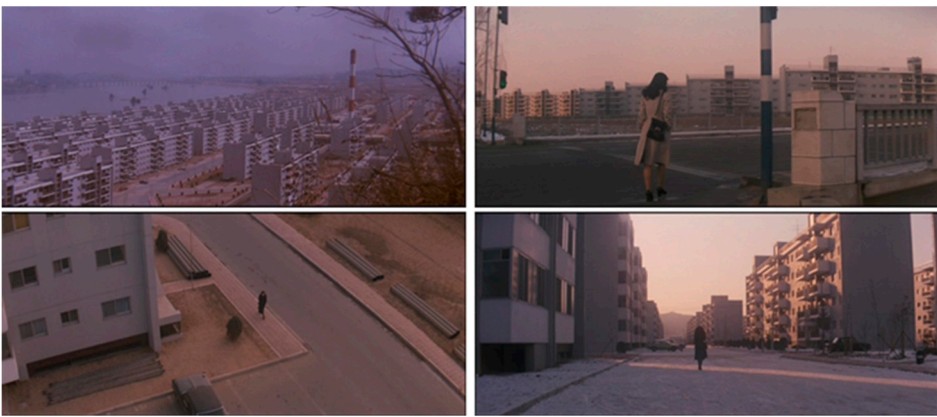

**Figure 1.** Scenes of the Banpo Housing Complex in "Night Voyage" (1973).

Hyun-joo, the owner of the trapped soul, represents the life of the citizens of Seoul in which the life of monotonous, repetitive, and uncharacteristic nature of citizens is expressed by illuminating the apartment complex on the screen. As shown in Figure 1, the seemingly endless apartment buildings and the procession of concrete form a barren urban landscape contrasted with the figure standing in it. As Valerie Gelezeau states, it resembles a skyline that is created by the dense towers, military barrack, matchbox, and military base [10].

The Banpo Housing Complex also appears as the main background in the film version of In-ho Choi's best-selling novel, "Heavenly Homecoming of the Stars" (Jang-ho Lee, 1974). It is the residence of the economically middle-class male protagonist, Moon-ho. Moon-ho's space is set in the Banpo Housing Complex's Building 35, and the meaning of the apartment space can be thought of from the perspective of Moon-ho and Kyung-ah. The male protagonist, Moon-ho, is a lumpen and lives off money from home. His life consists of sleeping, eating, washing, painting, and then sleeping and eating again in the apartment. In terms of his everyday life, Moon-ho is not inconvenienced by living alone, but the apartment is reflected as a boring and lonely space except for the periods that he is with Kyung-ah. Even during these periods, the apartment is still represented as an empty space except for the earlier happy times. For Moon-ho, the apartment space is the lonely shelter of a lumpen, where everyday living continues without interaction with the neighbors. For Kyung-ah, Moon-ho's apartment is a refuge rather than a haven. Kyung-ah, who used to be clean and innocent without any troubles, has gone through several men as she carved out her career in the society and has fallen into a degenerate lifestyle. In this state, she reaches Moon-ho's apartment. In line with Kyung-ah's wish to have children and live happily, the scene of children playing in the playground of the apartment complex often appears. For Kyung-ah, Moon-ho's apartment is a refuge for happiness, and at this time, the apartment space symbolizes sympathy and salvation in her dead-end life. Unfortunately,

this refuge, sympathy, and salvation are never realized, and Kyung-ah ends her life. Figure 2 shows the scenes of the apartment as an unfinished refuge and the lonely haven of a lumpen.

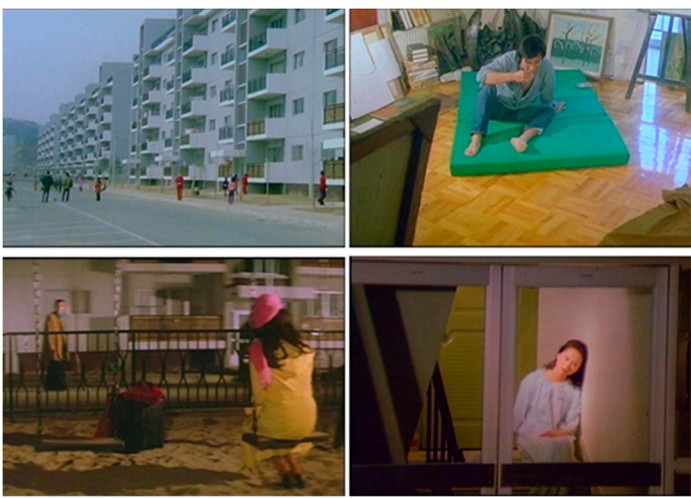

**Figure 2.** Scenes of Moon-ho's apartment in "The Heavenly Homecoming of the Stars" (1974).

In "Winter Woman" (Ho-sun Kim, 1977), a 38-year-old divorced man, Heo-min, appears alongside the main female character, Yi-hwa. The place in which Heo-min lives alone is a luxury apartment located on the banks of the Han River. This apartment was in the Jumbo apartments that residents moved into in 1974 (Ichon-dong, Yongsan-gu, Seoul; 1 building and 144 units). At that time, it was a high-rise 18-story apartment. As shown in the top right side of Figure 3, the interior of the apartment is drawn richly, and is reflected in a scene that emphasizes the vertical height of the apartment and the view of the South Han River. There is also the Dongbu Ichon-dong apartment (a 165 m$^2$ mansion completed in 1971) opposite the apartment. However, the content of the scenes at the bottom of Figure 3 includes the following expressions: Heo-min watches Yi-hwa with a very empty face, and Heo-min enters from the veranda feeling empty. His silhouette looks lonely while he smokes a cigarette [11].

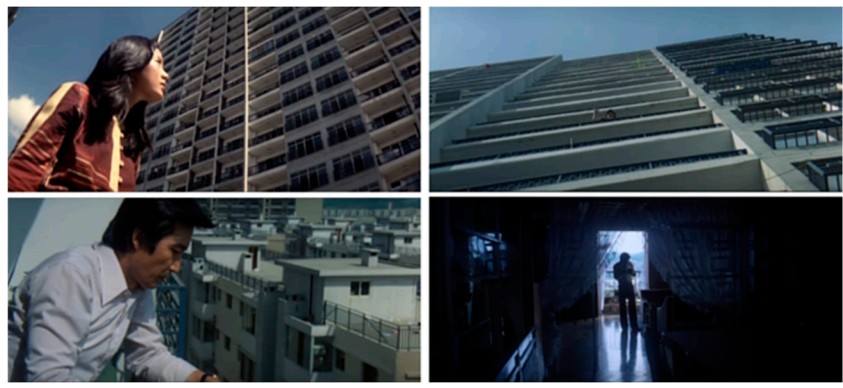

**Figure 3.** Scenes of Heo-min's apartment in "Winter Woman" (1977).

In the end, Heo-min's apartment space, described as an ordinary living environment of the middle class, is revealed to be personifying loneliness and emptiness itself. This loneliness and psychological instability of a single man continues to the end of the film. In the 1970s, the middle classes were realized through apartments, and there were the phenomena of the image formations of high-end apartments in Gangnam, Seoul. Nevertheless, this positive image of the apartment was not revealed through the film. As seen in the three films explored above, the apartment turns out to be a place of loneliness and anonymity, or an inconvenient living space for people on low incomes. This is partly

due to its relation to social conditions such as the opening of the Revitalizing Reforms era in 1972, the blind industrialization, and the degeneration and recession of the film world.

In "The Ae-ma Woman" (In-yeob Jeong, 1982), modern high-rise apartments along the Han River appear as the residential space of the main female character, Su-bi Oh. Su-bi, who has been living a financially rich life with her husband and young daughter in a luxury Western-style house with a garden, but leaves her daughter in her husband's house and moves to a luxury apartment along the Han River alone when her husband suddenly goes to prison. The apartment is set up as a space for single women with active consciousnesses. This contrasts with the house where the family lived together, and also with her mother-in-law's two-story house. Meanwhile, Su-bi accidentally meets Dong-yeob, a young art student. The rural house of Dong-yeob is filthy but has a warm and cozy atmosphere. The apartment chosen as the living space of Su-bi in the film symbolizes a temporary space of a single person when compared to the detached house, which symbolizes a family space. It functions as a space of coolness, emptiness, and loneliness compared to the humane and warm country house.

Su-bi, who lives away from family for the first time since her marriage, lives as a woman and not as a wife or mother for a while. She is materially rich but needs peace of mind and healing. Therefore, an appropriate level of anonymity is needed through the space of her apartment. However, when Su-bi discovers that Moon-oh, a former lover, lives upstairs, she suffers a serious blow to her privacy and eventually experiences greater mental confusion and conflict. This is because Moon-oh persistently demands a meeting with her by breaking onto her veranda. Her apartment life, which was calm for a moment, is in crisis when her anonymity is broken. The apartment was chosen by the single woman because of its convenience and anonymity. This phenomenon is also reflected in "The Carriage Running into the Winter" (So-young Jung, 1981). In this film, the camera zooms in and out to illuminate the small residents in the huge facade of the high-rise apartment, as if to highlight the lonely human beings in a big city. The apartment is also expressed in a uniform, repetitive image to emphasize the anonymity, akin to the scenes in Figure 4.

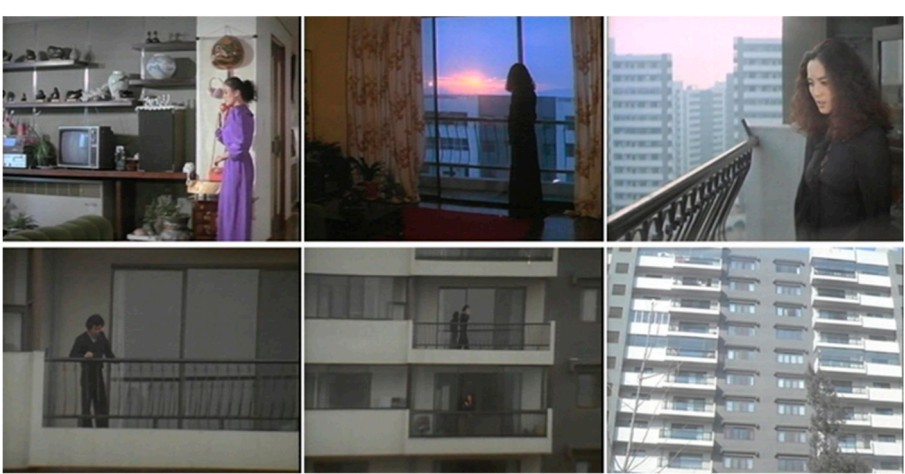

**Figure 4.** Images of emptiness and loneliness in "The Ae-ma Woman" (1982).

The film of the original work by O-young Lee, "Three Times Each for Short and Long Ways" (Ho-sun Kim, 1981), unfolds a story featuring the repetition and anonymity of everyday life without any change and uses modern people as its main subject matter. The space selected to visually reveal the repeatability and anonymity is a high-rise apartment complex in Gangnam, Seoul, and the Eunma apartments in Daechi-dong (1979). There is a scene in which the protagonist, a typical office worker Jong-sil (37 years old, Effect Man), complains about his boring and repetitive daily life and the stuffiness of his apartment while coming home from work. This is shown in the following transcript from the film:

*S# 12. Apartment entrance hall (night):*

[Jong-sil stumbles as he is drunk]

Jong-sil: When my cuckoo clock makes a cuckoo's call exactly eight times at eight o'clock in the morning, I head to work by suffering in a jam-packed bus. The cuckoo starts to announce seven o'clock in the evening.

[Jong-sil heads toward the elevator]

When a cuckoo's call is made, I leave work with my tired body and the same face and the same bag as in the morning. When a cuckoo's call is made, I go to work, and when a cuckoo's call is made, I leave work.

*S# 17. Mi-ah apartment (night):*

Jong-sil: As for today, boredom is over! Let's leave! I'm moving! I hate the apartment! It's over! This apartment is like a chicken coop! [12].

Jong-sil, who invents a brilliant sound effect, receives a blank check from an Arab company. Jong-sil gets drunk to celebrate that night and goes in an elevator. He does not go to his own house, suite 805, but to suite 905, the residence of a 28-year-old girl, Mi-ah, on the floor above. Jong-sil is unintentionally trapped there and watches on TV how his disappearance is distorted and spread by stakeholders and public opinion. Jong-sil thinks about human value and identity, and in the end, hears the news that he is dead. Finally, to restore himself, Jong-sil leaves the apartment and heads to the church where his funeral is being held. In the bottom right side of Figure 5, the scene of Jong-sil running through the apartment is symbolic as it is significant that the city dweller, disillusioned with his boring and repetitive routine and frustration over his henhouse-like environment, must escape from the high-rise apartment concrete jungle for his own recovery and rebirth.

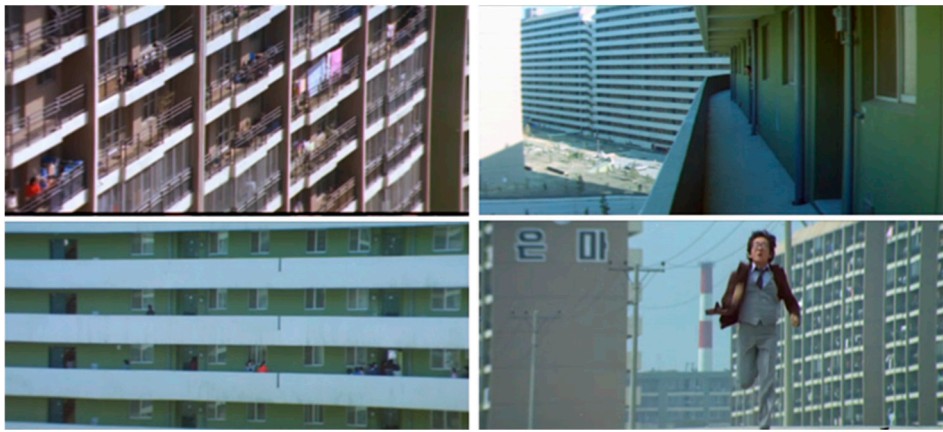

**Figure 5.** Images of repetition and anonymity of the Eunma apartments in "Three Times Each for Short and Long Ways" (1981).

In line with the compact growth of the Korean economy, the quantity-based apartment propagation policy that began in earnest in the late 1960s has allowed busy urban people (living in apartments) to enjoy physical conveniences and personal comforts but has led to further social isolation and a loss of wider communal open spaces. This is a cultural phenomenon caused by social structures in the process of industrialization. The apartment residents have isolated their territories to only having relationships with nuclear family members within their closed housing spaces. Therefore, living in an apartment most typically reveals the tendency to individualize modern society, and reveals the loss of community life as the place that connects them [13].

The characteristic of the original film and screenplay of In-ho Choi, "The Flower at the Equator" (Chang-ho Bae, 1983), is that it reflects the disorientation and wasted affection of modern people through a film that depicts modern urban peoples' distorted cultural landscapes of affection [8]. Through the story of the male and female main characters, the film addresses the loneliness and

alienation of urban people. To represent this context, large high-rise apartments in the center of a city are chosen, which have an image of anonymity and closure. These are the Sin-Hyundai apartments (1983; 1,924 units) in Apgujeong Gangnam, Seoul, which allowed residents to move in at the time of shooting. The male protagonist, "Mr. M," appears, who is a lazy and unsociable free timer in his twenties. He lives alone in a high-end apartment on the money that his father provides him. "Mr. M," who has been killing time doing nothing, sees a woman in the opposite apartment through a telescope. The woman, who also lives alone, is Sun-young Oh. In the film, there are many scenes related to loneliness and alienation that the protagonists feel in their apartment spaces. The film, meanwhile, treats the beginning and end scenes in the same way by faintly illuminating the searing city landscape. The bottom of the cityscape is filled with gray concrete. That is, the barrenness of affection, loneliness, and alienation that modern urban people embrace is represented by the shimmering concrete and huge apartment complex in the center of the city. Related scenarios are as follows.

S#01. Apartment

(Symbols of urban civilization in the 1980s: Our country. Desert's mirage. Jail, inanimate object.)
(Situation in the wind blowing sand: Searing apartment like a desert's mirage)
(Situation of a telescope: Downed window, minerals.) (Zoom-in into a black apartment window.)
(Move towards the back of the man sitting on a desk by the empty living room window in the apartment.)

S#03. Apartment

(Lost humanity: End of conversation and interest. Expressed M's whole loneliness, alienation, and idleness.)
(F.S) (marathoners filled like ants.) (S.T)
(Appearance of children in the playground like ants.) (Omit)
(Square of dawn: View of woman who ran out. Disconnected look. Inside the apartment.) [14]

The scene of the apartment glaring like a mirage in the desert is defined as a symbol of urban civilization while linking with the meaning of a prison, an inanimate object, the loss of humanity, and disconnection. Moreover, the barren city full of apartments is represented by a desolate desert. This is reflected in the monologue of "Mr. M" in the last scene of the film. As such, the film addresses the loneliness, alienation, and barrenness of affection of the middle class, and is based on the anonymity of the modern city like the main character's name, which is "Mr. M" ("Mr. M" changes to "Mr. Man" in the last scene of the film). Along with this anonymity, the closed living spaces are faithfully reproduced in the apartments, as shown in the scenes of Figure 6.

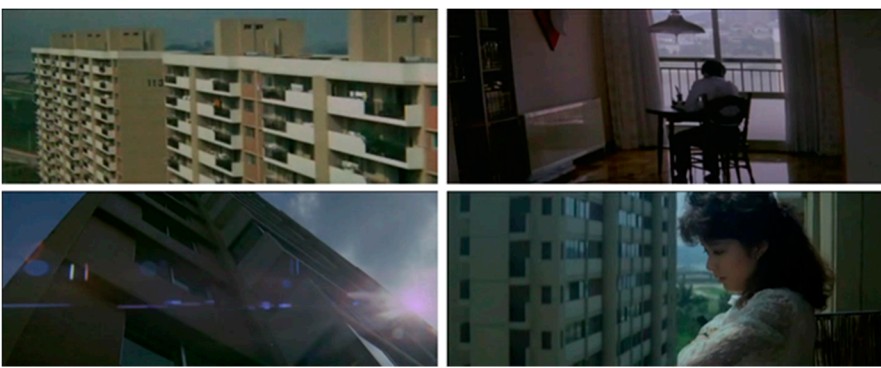

**Figure 6.** Anonymity and closure in the city in "The Flower at the Equator" (1983).

*4.2. Images of Inappropriate "Family Spaces"*

Although the apartment complex along the Han River was aimed at the middle class in the 1970s, it is hard to find an apartment that was set as a middle-class family space in the films of the

1970s. This suggests that apartments were not sufficient to be recognized as a housing space for a family (especially for large families) at the time. As a suitable example, we examined "Byung-tae and Young-ja" (Gil-jong Ha, 1979). In the original scenario of "Byung-tae and Young-ja" (original work and dramatization by In-ho Choi), the apartment is set as a space where a harmonious five-person family lives. This is unusual in the light of the cases in the previous periods. The appearance of a healthy and intimate family of a male protagonist (Byung-tae) in his twenties, his mother, elder brother, elder brother's wife, and younger brother blends in with the residential environment of the modern apartment. Ultimately, the apartment in the scenario is defined as a space that contains a harmonious middle-class family in modern society. However, this apartment does not appear in reality in the film. This space of the five-member family is replaced by an improved *hanok* (traditional Korean house) with a yard. As such, "S# 68: Byung-tae's apartment" and "S# 71: Byung-tae's apartment square" in the original scenario are changed to "S# 68: Byung-tae's house" and "S# 71: In front of Byung-tae's house" in the script. Ultimately, the space appears as a hanok with an alley, gate, and yard in the film, as shown in Figure 7 (which concerns the scene where Byung-tae introduces his girlfriend to his family). Considering the social contexts at the time, it can be said that the apartment was not sufficiently recognized as a harmonious, large family space, and that a hanok was more suitable as a relatively harmonious family space than an apartment.

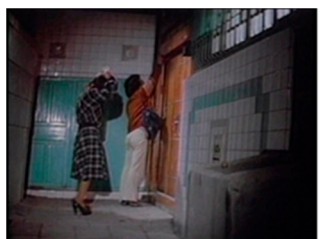 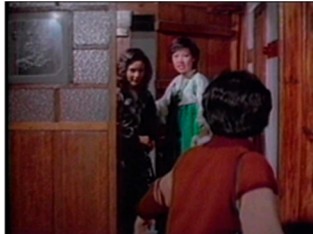 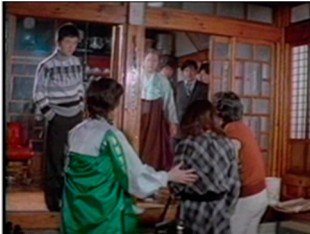

**Figure 7.** Scenes of an improved hanok with a yard in "Byung-tae and Young-ja" (1979).

A similar example is a film "The Oldest Son" (Doo-yong Lee, 1984), which addresses the breakdown of the traditional family system and filial piety. In this film, the difficult and negative perspective of apartment living for an elderly couple is revealed. As the hometown is chosen as a submerged district near a reservoir, the elderly parents come to the capital after living as ordinary villagers for the majority of their lives. In Seoul, they have three sons and two daughters. The elderly parents are not comfortable with suddenly living in a row house (their oldest son's house), and move to a temporary house near a large open plot for a while. They then move again to an apartment, which they lease through a deposit. Their apartment life is not easy, and the aged mother eventually ends her life in the apartment as she cannot adapt to life in the city.

In the scenes of "The Oldest Son," many apartments are described as being a sea of concrete. The aged mother feels dizzy in front of the apartment building. She says that she gets goosebumps as she feels she is being sucked into a black coffin. It is a completely different atmosphere from the situation where she planted a small farm in the corner of the empty plot at the temporary house and had various vegetables and flowers. To maximize the difference between the yard environment (which enables elders to have a pastime) and the apartment life (which alienates elders), the leased house was set on the highest floor of the apartment building.

The top left-side of Figure 8 shows a scene in which the elders look at the road's exclusive use of motor vehicles from the second floor of the row house. The bottom left side of Figure 8 shows the elders first arriving at the apartment and looking up at the building. The bottom right-side shows the coffin being brought down by a large rooftop crane upon the death of the aged mother.

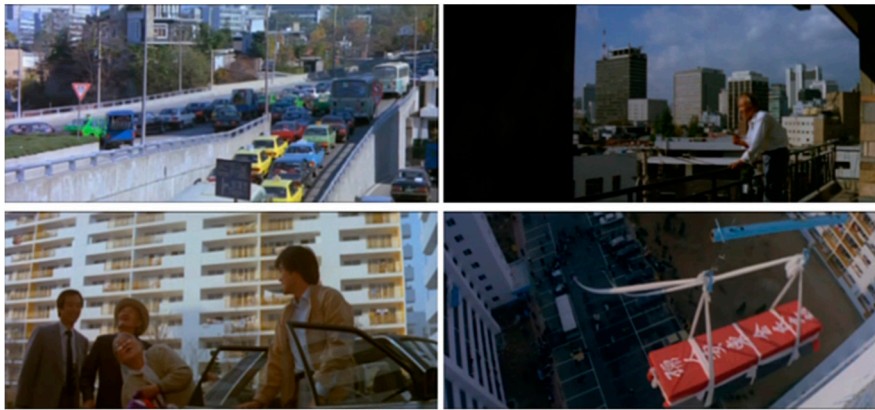

**Figure 8.** Scenes of life and apartments in the city in "The Oldest Son" (1984).

*4.3. Side Effects from Large Scale Developments*

The period between the late 1980s and the mid-1990s was a time when the housing market grew with the economic boom in Korean society. It can be regarded as the maturity of the so-called urbanization with the development of large-scale housing sites and the intensification of competition for goods. Further, densification via high-rise apartments had begun, and new town constructions, urban redevelopments, and reconstruction expanded. However, there were side effects that aggravated the urban environment, including the loss of places of life and memory, the dissolution of family, the intensification of relative poverty, and the formation of a desolate landscape. The following two films relate to these aspects.

"A Dwarf Launches a Little Ball" (Won-se Lee, 1981) is a film that focuses on the suffering and sorrow of life that poor and marginalized people weave. The main narrative of the story is a conflict over the occupancy rights of an apartment that a family receives, who live a difficult life in an illegal shack, along with a voluntary removal order from the landowner. Here, the right to move into the apartment does not serve as a new life opportunity for the poor, but rather a cause of loss of life and the dissolution of the family. For the rich, it is depicted as an object of speculation.

The meaning of the apartment at the time or the purchase right for the apartment is identified through the film, which establishes the mid-1970s as its temporal background. First, from the perspective of the economically stable real estate agent Woo-chul, the apartment is only the subject of real estate speculation, and its meaning as a living space is not reflected in the film. No attention is paid to the dark reality of the underprivileged as caused by the occupancy of the apartment. It is only worth as a tool for the agent's own benefit.

An apartment is an object that cannot be owned by poor people living in the shack. Woo-chul's high-rise apartment in the film appears as modern and clean, contrasting with the shack. Since a significant amount of money is required to live in such an apartment, the dream of buying their own house is impossible for the poor Bul-ee family. The right of occupancy was not for them in the first place. Tight demolition deadlines and contract dates forcibly notified them of such. Even if they made a contract and paid the down and middle payments, there is no place for them to stay for at least six months after the demolition. Eventually, the place of the family life with memories will be lost in exchange for receiving a small amount in the form of a relocation fee, or gaining the right to move into an apartment. Further, it leads to the dissolution of the family. As such, the right of occupancy comes as a medium to induce the family's dissolution, and is not ultimately recovered. This is because the father, Bul-ee, has already died, although his daughter, Young-hee, returns home with the family's apartment sale agreement.

In the early and mid-1970s through to the 1980s, during the housing construction boom, private-led housing dissemination policy and apartment complex development began in earnest, there was an economic downturn that resulted in sluggish housing construction and materialistic development.

Since 1987 and the economic boom, the housing market grew again. The film "Chilsu and Mansu" (Kwang-su Park, 1988) appeared at this time. The historical and social background of the time can be summarized as economic development, mass housing, Gangnam development, land development crazes, land speculation, and the establishment of the military regime. The film marks the beginning of the democratization era, and shows Chilsu and Mansu, who live a poverty-stricken life in the back alley of the city, as an expression of the universal Korean impression of modern Korean society.

Chilsu and Mansu paint apartments for living. They color apartments that they themselves cannot own or live in. Mansu's warehouse-like house in the city (Oksu-dong) always contrasts with the surrounding apartments (see the top left scene in Figure 9). Their complaints and anxieties about the unsolved reality accumulate and they shout for fun at the rooftop of a building (Figure 9: Top right side). They unwillingly confront the crowd on the ground, who misunderstand their actions as a demonstration or suicidal disturbance, and eventually spread to a social event as journalists and soldiers are dispatched. Although the rooftop where Chilsu and Mansu shout is a place that is physically open to the sky, it paradoxically symbolizes a closed, isolated space that is absent of communication and a place of disconnection. The main themes of the film are the social accusation through the happenings of the marginalized, the impossibility of communication among the classes, and the relative poverty caused by the Gangnam apartment complex and high-rise buildings. In order to effectively convey this message, apartment scenes frequently appear to make the apartment symbolic of the possessions of the rich and the flow of large mainstream society. Further, as a way of expressing this visually, the Sinbanpo apartment's image of vastness and compactness is used as shown in Figure 9, and the camera actively uses a bird's eye view and panning.

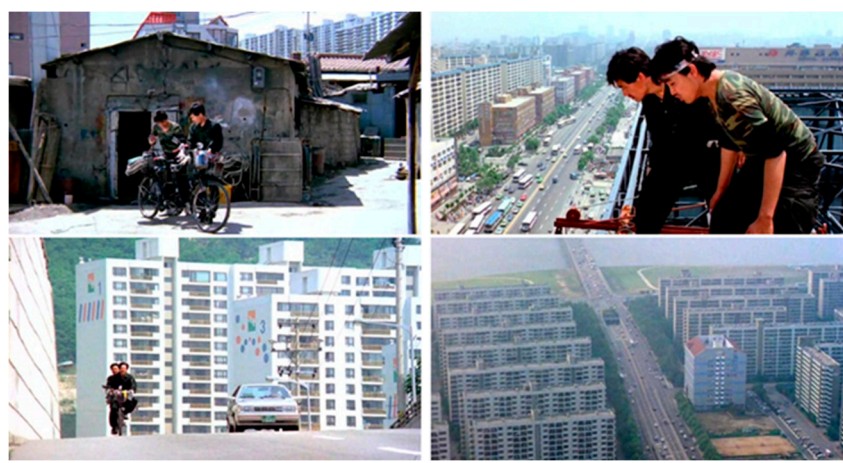

**Figure 9.** Apartment scenes in "Chilsu and Mansu" (1988).

The film "Green Fish" (Chang-dong Lee, 1997) was created on the basis of the social background of the change of place by the construction of a new city. The film focuses on the protagonist, Makdongee, whose identity is destroyed by the dissolution of place in the fallen urban world. Makdongee, 26 years old, who is just out of the military, is on his way home, driving down the Ilsan subway line. He loses direction from the station, and the only thing that has not changed in the town is the big tree in the yard of the house.

The leader of the changes in his hometown is the apartment in the new town. Here, the apartment is described as a "castle of light" with "gorgeous stage equipment," as well as having a "uniform and grim look" [15] (pp. 67–71). The apartment complex was the home of Makdongee's family. Now that the hometown has disappeared, family members are scattered. Makdongee's small dream is to run a restaurant with his family. The scene where this change in place is clearly visible is shown in Figure 10.

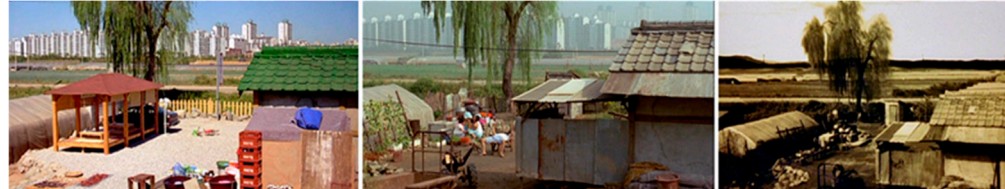

**Figure 10.** Changes in place and landscape over time in "Green Fish" (1997).

The changes by the development of the new town Ilsan means the loss of place and memory behind the urban industrialization and further symbolizes the dissolution of Makdongee's hometown, the fragmentation of his family, and a sign of the times. In the last scene of the film, Makdongee's family reunites and runs a big treehouse restaurant without Makdongee, who has died. This can be interpreted as the lost place is not recoverable, as time cannot be returned to the past.

*4.4. Place as a Lack of Communication, Misunderstanding, and Distrust*

Another negative view of apartments is the lack of communication, misunderstanding, and disbelief. This is an extension of the loneliness, anonymity, and closure mentioned earlier, and is a topic that stands out further as it becomes more relevant to modern times. In particular, in the films from the 1990s in which an apartment appears, the relationship, misunderstanding, and lack of communication between neighbors are addressed in the background of an apartment, such as in "A Hot Roof" (1995), but with the exception of the daily appearance of the middle and working classes. Among them, the breakdown in communication is also a topic that previously appeared in "Chilsu and Mansu" and "The Age of Success" (1988). In the scenario of "The Age of Success," the apartment is expressed as follows:

> *S# 134: Mansion apartment complex (at night)*
> The protagonist walks through the apartment buildings. The shadow of a mercury lamp draws a long and heavy shadow. The apartments feature walls of disconnection and division. The walls will not echo in that no one would know who laments [16].

The main stage of the film "301, 302" (Chul-soo Park, 1995) features a "new hope bio-apartment" as a cinematic setting that has been designed and constructed biotechnologically as a modern apartment located in a large city. Although the interior of the two female protagonists' apartments are uniquely presented to reflect their set personalities and tastes (Figure 11), the film focuses on their intimate loneliness and difficulties in communication. According to the script, both young women failed to get married, and live by themselves in suites 301 and 302. The apartment is set up as a space for modern single women. As shown in Figure 11, the interior of suite 301 shows the kitchen-oriented space, which is designed by the interior designer for Song-hee's requirements. There is an oversized table at the center of the open one-room space. The personality of chef Song-hee, who always pursues neatness and perfection, is revealed with a clean but cold metallic finish, sensuous furniture design, lighting fixture, etc. On the other hand, the interior atmosphere of 302 is contrary to that of 301, where warm-feeling wood was used throughout the floor and bookcase, and sunlight is flowing into the room rather than artificial lighting. Even through these bold and experimental interior settings, the film talks about the difficult communications between writer Yoon-hee in suite 302 who has anorexia nervosa and chef Song-hee in suite 301. This lack of communication, misunderstanding, and disbelief in apartments have continued to appear in recent works such as "Possessed" (2009), "The Neighbor" (2012), and "Hide and Seek" (2013).

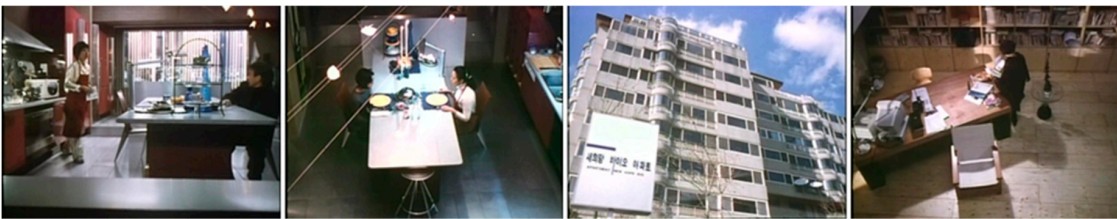

**Figure 11.** The interior of suite 301 (left) and suite 302 in "301, 302" (1995).

## 5. Discussion

This study was carried out on the premise that film inevitably reflects contemporary times, architectural cultures, and public thoughts and emotions. It examined the negative perceptions of apartments by mainly analyzing the films in which apartments appear between 1970 and 1999. The findings are summarized as follows.

In the 1970s and 1980s, apartments emerged as an urban middle-class space of loneliness, anonymity, and closure, as uniformly developed large-scale apartment complexes highlighted. Further, apartments were expressed as being unsuitable as a space for a large family, and its characteristic of alienating the elderly was highlighted. In the 1980s and 1990s, the negative aspects of apartment developments were highlighted. They included the loss of place (place of life) and memory, dissolution of family, deepening of relative poverty, and creation of standardized and desolate landscapes. Negative perceptions deepened in the 1990s when films mainly reflected the apartment's daily space and displayed the lack of communication between neighbors, their dismissals, misunderstandings, and their disbelief.

As mentioned in the introduction, exaggeration and distortion occurred in the expression of apartment culture in the films. This is a part that requires attention in interpreting accurate housing culture through films, and the two cinematic characteristics revealed in this study can be summarized as follows. First, film pays more attention to contemporary society than architectural culture. The films in the 1970s, including "Night Voyage" and "The Heavenly Homecoming of the Stars", focused on themes, such as loneliness and anonymity, rather than positive aspects of apartments that existed at the time, such as convenience and luxury living spaces. This is partly due to social situations such as the Restoration Age, blind industrialization, and the downturn of the film world. Second, film exaggerates a place's extreme emphasis on a character or story by its nature. Examples include the emptiness and loneliness in "The Ae-ma Woman," loss of identity in "Three Times Each for Short and Long Ways," and loss of humanity, closure, and alienation in "The Flower at the Equator." In addition, in "The Oldest Son," the apartment is referred to as a "dark coffin," and in "Green Fish," the irreparability of a place is connected to the death of the protagonist.

## 6. Conclusions

Based on the contents of this study, the following two issues can be discussed regarding the sustainable housing culture of Korean apartments. First, films made before the 1970s did not reveal the negative aspects of apartments. Ultimately, the large complex developments of the Han River mansion, Yeouido Sibeom apartments, and Banpo Complex were aimed at the middle classes. Since the images of loneliness, bleakness, anonymity, and closure in the films of that time were expressed as the feelings of the middle classes in their large, dense apartment complexes, the three elements of the complex, the middle class, and the negative perception of apartments can be seen as equally valuable concepts of the time. In short, the public's negative perception of apartments coincides with the construction of a "uniform large-scale apartment complex." Second, the negative perceptions intensified the concept of loneliness and anonymity to the loss of place, misunderstanding, distrust, and anxiety. Likewise, in recent years, especially in the films of the 2000s, there are hardly any positive perceptions of apartments such as longing and curiosity. There are several possible interpretations of this phenomenon

of the absence of positive perceptions and the deepening of negative perceptions. The first is social fatigue. The accumulation of sociocultural fatigue by the construction and dissemination of large-scale apartments, which have been running since the late 1960s, emphasizes only quantitative growth. That is, there is no reason to regard an apartment, which has become a representative housing type in Korea, as a new or interesting object in terms of its quantity, uniform housing environment, public perception, or value as a cinematic topic. Second is the economic aspect, that is, the aspect of the apartment value as real estate. When the IMF financial crisis broke the myth of the apartment's invincible price, and the supply of apartments was nearly saturated nationwide, apartments lost value as a means of multiplying property, which was one of its most important attractions. In fact, many of the positive aspects of apartments that were maximized in the films of the 1980s and 1990s were attributable to economic abundance, such as the joy of buying a home, rising status, and success. The final factor is the fundamental nature of an apartment that alienates people. Korean apartments were a result of compact economic growth and were "goods" that were mass-produced and forcibly disseminated by the new production system of modern industrial society. The user's personal preferences were not reflected, and the target was an ambiguous object of the "public" or an "unspecified majority." Accordingly, in terms of the residential field of modern Korean society, apartments were a place that almost nakedly reveals human alienation, despite it being the most common type of housing. The film genre identifies and presents such situations very sensitively and provides the reason why the architectural community, who must think about the inherent problems of housing constructions, must-watch films.

This study is significant in that it has explored apartment housing culture through films, which are a popular and visual medium from the perspective of the general public, rather than the perspective of an architectural expert. It attempts to expand the scope of architectural research through image media rather than print media (newspapers, advertisements, novels, and magazines). However, at the same time, establishing only a very limited cultural product known as the "image medium" as a subject of the analysis is a limitation of the study. In the future, it is hoped that this study can contribute to bringing newness to various fields within architecture through the architectural examination of popular films.

**Funding:** This research was supported by the Keimyung University Research Grant of 2017.

**Conflicts of Interest:** The authors declare no conflict of interest.

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
