# Peer review of "The Negative Perceptions of Apartment Culture as Represented in Korean Films during the 1970s–1990s"

_sustainability, doi:10.3390/su12073013_

Round 1

Reviewer 1 Report

The journal Sustainability concerns research on the environmental, cultural, economic, and social sustainability of human beings, and sustainable development. The manuscript appears to fit this brief through its consideration of sustainable living spaces (apartments) in South Korea using perspectives from popular films.

It is impressive that the novelty of the study appears to lie in its objective to blend humanities and architectural fields together to discuss negative perceptions of apartment spaces in Korean films.

One minor thing, according to the journal template, the journal requires the author to briefly highlight the principal conclusions in the introduction section. This part is missing and needs to be inserted in the latter part of the introduction.

Author Response

Point 1: One minor thing, according to the journal template, the journal requires the author to briefly highlight the principal conclusions in the introduction section. This part is missing and needs to be inserted in the latter part of the introduction.

Response 1: For the positive sustainability of Korean apartment culture, the main purpose of this paper is to examine and diagnose negative perceptions of apartments rather than to provide specific planning guidelines. These are mentioned in the latter part of the introduction.

Additionally, the followings are mentioned in the conclusion; The three elements of the complex, the middle class, and the negative perception of apartments can be seen as equally valuable concepts of the time. In short, the public’s negative perception of apartments coincides with the construction of a “uniform large-scale apartment complex.” Korean apartments were a result of compact economic growth, and were “goods” that were mass produced and forcibly disseminated by the new production system of modern industrial society. The user’s personal preferences were not reflected. The film genre identifies and presents such situations very sensitively and provides the reason why the architectural community, who must think about the inherent problems of housing constructions, must watch films.

Reviewer 2 Report

The writer makes a case for films being important for architects, and discusses a set of Korean films from the 1970s through 1990s to unpack understandings of apartment living. This is potentially very interesting within the context of Korea, given the rapid change from a largely agrarian economy in the 1950s to what is now one of the world's most dynamic and multi-facetted economies. Indeed, there would be some sense that China might learn from Korea's experiences with urbanisation and adoption of multi-family housing.

However, one gets a sense from the article that the viewpoint of the author shapes the reading of the films (or at least how the findings are communicated), rather than the reading of the films leading to the viewpoint. This is likely an issue of writing style rather than one of biased readings of the films themselves. Perhaps the brevity with which the films are discussed leads to this. One of two paragraphs per film tends to lead to similar conclusions for each of them: loneliness, anonymity, bleakness. It might make for a more fruitful approach to discuss fewer films, but in greater depth, with more vivid descriptions, in order to tease out deeper readings of them. For example, this reader would be very interested in hearing more about the "bio-apartment" discussed in lines 455-464, as it seems to align chronologically with the development of biotech in Korea, especially around cloning. More time spent on particular scenes or settings - their texture, lighting, materials, and spatial character - in the films would enable the readings of them to flow out of them, rather than being seemingly imposed on them. A reworking of the Results would make for a much more solid contribution

One also has to wonder about these films within the context of rising material wealth and promises of a better future. Was apartment living in these films merely 1/ expressing larger issues in a society shifting away from working the land to consumerism, and 2/ indicative of the commonality of living in multi-family surroundings? With more people living in apartments, were not apartments becoming the setting for the full gamut of human conditions? Is the apartment and apartment building perhaps also an actor in these films, in this sense?

Regarding the method, on line 97, it is unclear how the author gauges "excellent and popular films". Were these award-winners and box-office winners? Also, on lines 101-104, it is unclear what the author is trying to say. Please rewrite this. 

Regarding sources, the quoted scripts do not appear to be referenced; they should be. Also other key work on film and architecture ought to be mentioned, such as the work of Juhani Pallasmaa, Ed Keller, Dietrich Neumann, and Renée Tobe.

Regarding writing and style, the author should look at the oddities in spacing and punctuation throughout the paper and correct these (e.g. lines 97, 109, 168, 201, 229, 283, 321, 413, 442, 494, and 502). The article is largely well-written, and otherwise has no problems, apart from the odd word her and there, which could be picked up with a careful proof-read.

Author Response

Response to Reviewer 2 Comments

 Point 1: However, one gets a sense from the article that the viewpoint of the author shapes the reading of the films (or at least how the findings are communicated), rather than the reading of the films leading to the viewpoint. This is likely an issue of writing style rather than one of biased readings of the films themselves. Perhaps the brevity with which the films are discussed leads to this. One of two paragraphs per film tends to lead to similar conclusions for each of them: loneliness, anonymity, bleakness. It might make for a more fruitful approach to discuss fewer films, but in greater depth, with more vivid descriptions, in order to tease out deeper readings of them. For example, this reader would be very interested in hearing more about the "bio-apartment" discussed in lines 455-464, as it seems to align chronologically with the development of biotech in Korea, especially around cloning. More time spent on particular scenes or settings - their texture, lighting, materials, and spatial character - in the films would enable the readings of them to flow out of them, rather than being seemingly imposed on them. A reworking of the Results would make for a much more solid contribution.

Response 1: Thank you for the sharp and accurate comments.

The characteristics of Korean apartments, such as uniformity, anonymity, and closeness, are already well known through existing studies. But in this study, different from the existing architectural studies, I wanted to explore the public perceptions of apartments that were revealed in the video media, and I tried to look at them as objectively as possible.

In the latter part of the section 4.4, I added an additional description of the interior atmosphere and design elements of the “new hope bio-apartment”. However, I would like to say that the bio-apartment was not a real apartment in Korea, but a setup in the film. As shown in Figure 11(newly inserted), only the exterior of an ordinary apartment was a real. Bio technologies such as retinal/fingerprint scanners, automated machines, and so on, were never shown in the film.

Further, the paper wanted to focus on the public’s perception of residential culture, instead of focusing on cinematic setting and design elements. Potentially, the architectural spaces or settings in films are expected to be covered by the following research topics.

Point 2: One also has to wonder about these films within the context of rising material wealth and promises of a better future. Was apartment living in these films merely 1/ expressing larger issues in a society shifting away from working the land to consumerism, and 2/ indicative of the commonality of living in multi-family surroundings? With more people living in apartments, were not apartments becoming the setting for the full gamut of human conditions? Is the apartment and apartment building perhaps also an actor in these films, in this sense?

Response 2: Korean apartments were a result of compact economic growth, and were ‘goods’ that were mass produced and forcibly disseminated by the new production system of modern industrial society. The user’s personal preferences were not reflected, and side effects would inevitably have occurred, such as uniformity, human alienation, lack of communication between neighbors, and so on. I expected to see these side effects in the films, and I hope these will be fully considered when planning apartments in the future.

Point 3: Regarding the method, on line 97, it is unclear how the author gauges "excellent and popular films". Were these award-winners and box-office winners? Also, on lines 101-104, it is unclear what the author is trying to say. Please rewrite this.

Response 3: It is judged that the artistically and publicly renowned films in the selected collections reflect the social and housing culture at the time better than other films that had not been included, as the collections themselves had high standards for inclusion. Three populations were constructed. The film selection criteria of these three populations were mentioned, such as: the award-winning or nominated films at domestic and international film awards, important films to understand Korean society and culture, and so forth. Please refer to lines 97-107.

Point 4: Regarding sources, the quoted scripts do not appear to be referenced; they should be. Also other key work on film and architecture ought to be mentioned, such as the work of Juhani Pallasmaa, Ed Keller, Dietrich Neumann, and Renée Tobe.

Response 4: Reference 11, 12, 14 have been added to the quoted scripts. It is true that I and my research have been influenced by the books of Dietrich Neumann and Donald Albrecht; however, they were not mentioned as references, because they were not directly related to the Korean films.

Point 5: Regarding writing and style, the author should look at the oddities in spacing and punctuation throughout the paper and correct these (e.g. lines 97, 109, 168, 201, 229, 283, 321, 413, 442, 494, and 502). The article is largely well-written, and otherwise has no problems, apart from the odd word her and there, which could be picked up with a careful proof-read.

Response 5: Thank you for highlighting these inaccuracies. The paper has been checked by a professional proofreader and all spacing and punctuation errors have been corrected.